# COVID-19 and Autoimmune Liver Diseases

**DOI:** 10.3390/jcm11102681

**Published:** 2022-05-10

**Authors:** Annarosa Floreani, Sara De Martin

**Affiliations:** 1Institute for Research, Hospitalization and Healthcare, 37024 Negrar, Italy; 2University of Padova, 35131 Padova, Italy; 3Department of Pharmaceutical and Pharmacological Sciences, University of Padova, 35131 Padova, Italy; sara.demartin@unipd.it

**Keywords:** SARS-CoV-2, COVID-19, autoimmunity, autoimmune hepatitis (AIH)

## Abstract

SARS-CoV-2 infection can trigger autoimmune responses, either by a systemic hyperstimulation of the immune system or molecular mimicry (or both). We here summarize the current knowledges about autoimmune liver diseases (AILDs) and COVID-19, focusing on (a) the risk of SARS-CoV-2 infection in patients affected by AILDs and/or under pharmacological treatment with immunosuppressants; (b) the capability of vaccination against SARS-CoV-2 to trigger autoimmune responses in the liver; and (c) the efficacy of vaccines against SARS-CoV-2 in patients with AILDs. Although unconclusive results have been obtained regarding the risk of being infected by SARS-CoV-2, generally indicating that all patients with chronic liver diseases have the same risk, irrespective of the etiology, the use of immunosuppressants in patients with AILDs seems to be correlated to COVID-19 severity. Few cases of autoimmune hepatitis (AIH) after SARS-CoV-2 vaccination have been reported, all characterized by a complete remission upon steroid treatment, but further evidence is needed to demonstrate the causality assessment. Humoral responses have been observed in patients with AILDs upon vaccination. In conclusion, the link between SARS-CoV-2 infection and AILDs is far to be completely elucidated. In these patients, the use of immunosuppressants has been correlated to an increase of disease severity and lower levels of antibodies upon vaccination.

## 1. Introduction

During the early stages of COVID-19 pandemic, the European Association for the Study of the Liver (EASL) and the European Society of Clinical Microbiology and Infectious Diseases (ESCMID) published a position paper to provide guidance for physicians involved in the case of patients with chronic liver diseases [1]. Specific recommendations were given to patients with autoimmune liver diseases in advising against reducing immunosuppressive therapy to prevent SARS-CoV-2 infections.

In this contest, unravelling the relationship between COVID-19 and patients with autoimmune liver diseases is crucial since these patients are generally characterized by an increased risk of infections due to the iatrogenic effect of immunosuppressive agents and the pathogenic mechanism of the disease itself. The purpose of this review is to provide an overview on the risk of SARS-CoV-2 in patients with autoimmune liver disease, also adding insights on the current use of the vaccines against COVID-19.

## 2. SARS-CoV-2-Induced Autoimmunity

It has been hypothesized that SARS-Co2 may trigger autoimmune mechanisms in genetically predisposed subjects [2]. In particular, an increase in several inflammatory cytokines including tumor necrosis factor (TNF)α, interleukin (IL)-1, and IL-6 has been demonstrated in patients with COVID-19 [3].

A recent review focused on the SARS-CoV-2 as an instrumental trigger of autoimmunity [4]. In general, three primary groups of factors (i.e., the ability of this virus to overstimulate the immune system, the formation of excessive neutrophil extracellular traps with related inflammatory responses, and the molecular resemblance between self-components of the host and the virus) can lead to hyper-stimulation of the immune system when varying from their physiological effect. These factors may contribute to the development of autoantibodies and autoimmune diseases. COVID-19 can activate a hyperstimulation of the immune system or, through the exposure to foreign peptides homologous to human peptides (molecular mimicry), contribute to the development of autoantibodies and autoimmune liver diseases. A list of numerous autoantibodies has been detected in patients with SARS-CoV-2 infection, including ANA (anti-nuclear antibody), ANCA (anti-neutrophil cytoplasmic antibodies), LAC (anti-phospholipid antibodies), etc. These autoantibodies have been generally shown in randomly chosen severely ill COVID-19 patients with no history of autoimmunity. As regard the autoimmune conditions linked to COVID-19 infection, besides the olfactory manifestations, several autoimmune diseases including SLE (systemic lupus erythematosus), myasthenia gravis, and systemic sclerosis have been described [4]. The exact mechanism by which SARS-CoV-2 leads to hyposmia/anosmia has not been elucidated so far. One hypothesis is a damage caused by the virus to the olfactory pathways [5]. Furthermore, olfactory manifestations have already been described in autoimmune diseases, including SLE, myasthenia gravis, and systemic sclerosis [6]. Interestingly, ANA, LAC, and ANCAs have been tested in 33 consecutive patients with COVID-19, 94% of whom had interstitial pneumonia [7]. Fifteen patients tested positive for at least one antibody (45%); four out of eleven tested positive for ANA had nucleolar staining, a reactivity that usually is present in diffuse systemic sclerosis. Of note, the presence of these subtype of ANA was associated with a bad prognosis [7]. However, the current literature on COVID-19 has focused mainly the development of autoimmune rheumatologic diseases [8].

Histopathological signs of autoimmune reactions have been observed analyzing autopsies from patients deceased from COVID-19 [9]. Diffuse infiltration of the lung, along with focal infiltration of the kidney, liver, intestine, adrenals, pancreas, and pericardium have been described. The infiltrate was dominated by T lymphocytes (CD3^+^ and CD8^+^ suppressors), suggesting that an autoimmune process might play a role in tissue damage.

Regarding liver diseases, a case of a female patient who developed primary biliary cholangitis (PBC) concomitant with Guillain-Barrè syndrome during severe acute respiratory SARS-CoV-2 infection has been reported [10]. PBC developed in a 44-year-old obese and hypertensive woman who was already suffering from Hashimoto’s thyroiditis. She had a severe course from a bilateral interstitial pneumonia and was treated in the intensive care unit. During her hospitalization, she developed an important increase in the cholestatic enzymes and was tested positive for ANA 1:160 and anti-mitochondrial antibodies (AMA) 1:640. Her histology was compatible with early PBC.

Furthermore, a case of autoimmune hepatitis (AIH)/PBC overlap syndrome has been reported in a 57-year-old man with a medical history of hypertension, pre-diabetes, and beta-thalassemia minor who suffered from COVID-19 infection with moderate respiratory symptoms in April 2020 [11]. This patient suffered from fatigue and arthralgias and was diagnosed with AIH/PBC overlap syndrome based on elevated liver enzymes, hyperbilirubinemia, SMA (smooth muscle antibodies), AMA positivity and anti-double-stranded DNA antibodies (anti-ds DNA). Anti-ds DNA positivity was reported in 38-60% of patients with AIH/PBC overlap syndrome [12,13,14]. Interestingly, the concomitant positivity for anti-ds DNA and AMA seems highly specific (98%) for the diagnosis of PBC-AIH OS [13].

## 3. Is the Risk of SARS-CoV-2 Infection Increased in Autoimmune Liver Disease Patients?

The relationship between autoimmune diseases and SARS-CoV-2 is very complex and only partially understood. An interesting study evaluating the association between autoimmune diseases and risk of COVID-19 has been performed in the population of Milan, an area particularly damaged by the virus, which includes the municipality of Codogno, where the Italian outbreak started [15]. A total of 20,364 test-positive and 34,697 test-negative subjects were analyzed. The adjusted odds ratio (OR) of having an autoimmune disease in COVID-19 positive-subjects compared to negative-subjects was 0.86 (95% CI 0.76–0.96). Comparing COVID-19 positive-subjects to a random sample of controls from the general population the adjusted OR of having an autoimmune disease was 0.98 (95% CI 0.90–1.08). In other words, patients with autoimmune diseases did not carry a specific risk for developing SARS-CoV-2 infection.

Moreover, data regarding the clinical presentation and outcome of COVID-19 in patients with autoimmune liver disease are limited to international registry studies and to retrospective case studies. During the first outbreak of COVID-19, a phone-based survey using a 26-query questionnaire to explore the clinical features of SARS-CoV-2 infection in patients with autoimmune liver disease under immunosuppression has been performed in Northern Italy [16]. Of the 148 eligible patients, 47 (32%) were children diagnosed with AIH or autoimmune sclerosing cholangitis; 101 (68%) were adults diagnosed with AIH (n = 97), primary sclerosing cholangitis/AIH overlap syndrome (n = 2) and PBC (n = 2). Thirty-nine patients (26%) with fever or respiratory symptoms were suspected COVID-19 cases. None required admission to hospital, and none discontinued immunosuppression. Four patients had confirmed COVID-19 cases (all had AIH under immunosuppression). One of them (a 23-year-old female with Trisomy 21) died from septic shock unrelated to COVID-19. The same incidence of COVID-19 in autoimmune liver disease patients was observed in the general population. Since the outcome was favorable in most cases, the authors suggested that tapering or withdrawing immunosuppressive treatment is not required.

Furthermore, another study conducted on telemedicine in Northern Italy during the COVID-19 pandemic in patients with autoimmune liver disease showed quite low rates of symptomatic SARS CoV-2 infection, with overall favorable outcomes [17].

Likewise, patients with scleroderma, evaluated with a phone-based survey, seemed not have an increased risk for developing severe COVID-19 [18].

Interestingly, the first case series of 10 Italian patients with AIH under immunosuppressive therapy has been described during the first outbreak of SARS-CoV-2 infection [19]. Eight patients were on biochemical remission. All patients had a respiratory syndrome; in 7 patients the dosage of immunosuppressive medication was reduced. In general, the clinical outcome was comparable to the reported cases occurring in non-immunosuppressed subjects.

Another retrospective international study analyzed the outcome of 110 patients with AIH from 34 centers in Europe and America [20]. Use of antivirals was associated with liver injury (*p* = 0.041), while continued immunosuppression during COVID-19 was associated with a lower rate of liver injury (*p* = 009). Cirrhosis was an independent predictor of severe COVID-19 in patients with AIH (*p* = 0.001). However, patients with AIH were not at risk for worse outcomes with COVID-19 than other causes of chronic liver disease.

Finally, data for patients with AIH and SARS-CoV-2 infection were combined from three international reporting registries and compared to those in patients with chronic liver diseases of different etiologies and to patients without liver disease [21]. The group with AIH included 70 patients, 83% of whom were taking immunosuppressive therapy. There were no differences in rates of major outcomes between patients with AIH and non AIH. Factors associated with death within the AIH cohort included age and advanced liver disease (Child B and C) but not immunosuppression.

## 4. The Impact of Immunosuppressive Therapy on Viral Infection

It has been hypothesized that the iatrogenic effect of immunosuppressive therapy can complicate the outcome of SARS-2 infection [22]. Corticosteroids inhibit the immune response and delay the clearance of pathogens, mainly suppressing the host inflammatory response which in the case of viral infections of the respiratory tract is the major responsible for lung damage and occurrence of acute respiratory distress syndrome [23].

The impact of AIH medications, including glucocorticoids, thiopurines, mycophenolate mofetil (MMF), and tacrolimus on the risk of worse COVID-19 severity has been evaluated in a large international retrospective study carried out between 11 March 2020 and 15 May 2021 [24]. A total of 254 patients with a median age of 50 years (range 17–85) with AIH were included, 92.1% of whom were on treatment with glucocorticoids (n = 156), thiopurines (n = 151), MMF (n = 22) or tacrolimus (n = 156), alone or in combination. Overall, 94 (37%) patients were hospitalized and 7.1% died. Baseline treatment with systemic glucocorticoids or thiopurines prior to the onset of COVID-19 was significantly associated with COVID-19 severity.

## 5. Can the Vaccination against SARS-CoV-2 Trigger Autoimmunity?

The first two approved mRNA vaccines included the BNT 162b2 Pfizer-BioNTech vaccine and the Oxford University/Astra Zeneca vaccine, ChAdOx1 nCOV-19 (AZD 1222). During 2021, sporadic cases of thrombotic events in young women following vaccination with AZD 1222 have been reported, particularly of the rare condition of cavernous sinus thrombosis [25]. Studies have documented a pathogenic mechanism due to the antibodies to the SARS-CoV-2 spike protein that cross-react with platelet factor 4 (PF4/CXLC4) in a similar way to autoimmune heparin-induced thrombocytopenia [26]. Furthermore, anecdotal cases of different types of autoimmunity triggered by vaccines have been reported. For example, a subacute cutaneous lupus erythematosus induction ten days after receiving the SARS-CoV-2 mRNA vaccine second dose (Pfizer, Cominarty) has been shown in a patient with primary biliary cholangitis [27].

Regarding liver autoimmunity, Table 1 reports the cases of AIH developed after vaccination against SARS-CoV-2 published in the recent literature [28,29,30,31,32,33,34,35,36]. A total of 9 cases have been described (7 females, 2 males) with an age ranging between 35 and 76 years. Two subjects had a concomitant autoimmune disease (Hashimoto’s thyroiditis). In five cases the trigger vaccine was the mRNA-1273 (Moderna), in three the PNT 162b2 Pfizer BioNTech, and in one case the ChAdOK1 nCOV-19 Astra Zeneca. Eight patients developed jaundice with total serum bilirubin ranging between 1.5 to 12 times the upper normal value (UNV). Serum alanine transferase was markedly elevated in seven patients (ranging between 579 to 2001 IU/L) and moderately increased in one subject (3 × UNV). All but one developed non- organ-specific autoantibodies. Interestingly, the autoimmune liver serology of one patient [29] was remarkable. While indirect immunofluorescence showed a pattern compatible with AMA, molecular testing for PDC-E2, OGDC-E2, and BCOADC-E2 was negative, indicating that AMA was different from the typical AMA of PBC. Moreover, sp-100 and gp-210 tested negative in immunoblotting. All subjects underwent liver biopsy and liver histology was typical or compatible for AIH. All patients were treated with immunosuppressive therapy followed by remission within five to six months.

The reported cases of AIH followed by vaccines suggest an AIH-like pathogenesis rather than a classical AIH. The majority of cases have a short latency time from vaccination to the onset of symptoms Moreover, liver histology does not describe severe fibrosis or cirrhosis which, otherwise, is present in 21–84% of cases in classical AIH at onset [37]. In classical AIH ANAs are present in up to 75% of cases (with or without anti-smooth muscle antibodies). However, ANAs can also be detected in fatty liver disease, drug-induced liver injury (DILI) or viral hepatitis. The pattern of ANA in AIH is often homogeneous or speckled [38]. More specific for the diagnosis of PBC are extractable nuclear antigens sp-100 and gp-210 which result in a typical pattern of “multiple nuclear dots” and “nuclear rim”, respectively, in immunofluorescence on Hep-2 cells [39,40]. Interestingly, three patients with AIH after vaccines were older than 70 years, and it should be stressed that AIH not rarely occurs in the elderly [41].

Amelioration under steroids which can be discontinued without relapse is the main characteristic of DILI AIH-like. For the published cases associated with COVID-19 vaccine only a long-term follow-up will allow to establish if this AIH is really a drug-induced liver injury. The mechanism of induction of autoimmunity by vaccine is unknown, but probably several factors may play a role, including genetic susceptibility and molecular mimicry.

## 6. SARS-CoV-2 Vaccination Response in Patients with Autoimmune Liver Disease

A multicenter, prospective cohort study (OCTAVE trial) examined the SARS-CoV-2 vaccine responses in patients with compromission of the immune responses (40% requiring hemodialysis, 21.8% with inflammatory bowel disease and solid cancer, 5.36% with hematological malignancies, 28% with inflammatory rheumatic disease, and 6.17% with hemopoietic stem cell transplant recipients [42]. This study demonstrated that 89% of patients seroconverted within four weeks post second dose whereas 11% of patients failed to reach seroconversion. A prospective observational study comparing the humoral and T-cellular immune response to SARS-CoV-2 vaccination in patients with autoimmune liver disease has been recently published [43]. One hundred and three patients with AIH and 125 patients with PBC or PSC were included. In almost all patients, a humoral vaccination response could be detected. Antibody levels were lower in AIH patients under immunosuppression compared with those without immunosuppression. Antibody titers significantly declined within seven months after the second vaccination. In the autoimmune assay of 20 AIH patients, a spike-specific T-cell response was undetectable in 45% despite a positive serology, while 85% of the PBC/PSC patients demonstrated a spike-specific T-cell response. Along this line, AIH patients seem to have an increased risk in acquiring SARS-CoV-2 infection while patients with PBC or PSC might be protected from the infection.

## 7. Conclusions

There is a rationale supporting the link between SARS-CoV-2 infection and AILDs, although the mechanisms regulating this relationship are far to be completely elucidated. A number of clinical observations evaluated the prevalence of infection and the severity of symptoms in patients with AILDs, demonstrating that the risk of infection is similar in all chronic liver diseases, irrespective of the etiology, but the use of immunosuppressants (steroids and thiopurines) has been correlated to an increase of disease severity, and also to lower levels of anti-SARS-CoV-2 antibodies upon vaccination. A limited number of cases of AIH has been reported after vaccination against SARS-CoV-2. All the cases have been successfully treated with steroids and did not relapse upon drug discontinuation. Furthermore, the causality assessment needs to be demonstrated by a long-term follow up of the patients. Further mechanistic studies are needed to describe in detail the role of an incorrect autoimmune response in COVID-19 patients and demonstrate the impact of SARS-CoV-2-mediated autoimmune reactions in the generation and management of liver autoimmunity.

## Figures and Tables

**Table 1 jcm-11-02681-t001:** AIH after SARS-CoV-2 vaccination.

Sex, Age	Pre-Existing Morbidity	Type of Vaccine	Time Onset after Vaccination	Clinical Symptoms	AutoAbs	Liver Biopsy	Treatment	Outcome	Ref.
F, 76	Hashimoto thyroiditis, urothelial ca.	mRNA-1273 Moderna	3 days after dose I	Jaundice	ANA 1:1280 ASMA 1:1280 pANNA 1:1280	Typical for AIH	Steroids	Remission	[28]
M, 63	Type II diabetes Ischemic heart disease	mRNA-1273 Moderna	7 days after dose I	Jaundice Fatigue	ANA 1:640; AMA; anti-gastric parietal cells 1:320	Typical for AIH	Prednisone	Remission	[29]
F, 41	Premature ovarian failure	mRNA-1273 Moderna	7 days after dose II	Jaundice	ANA 1:80 ASMA 1:40 SLA+	Typical for AIH	Prednisone	Remission	[30]
F, 80	Hashimoto thyroiditis	BNT 162b2 Pfizer BioNTech	7 days after dose III	Jaundice	ANA 1:160 speckled	Typical for AIH	Prednisone	Remission	[31]
F, 35	Gestational hypertension 3 months before	BNT 162b2 Pfizer BioNTech	7 days after dose I	Jaundice	ANA 1:1280 homogeneous; DNA Abs 1:80	Drug/toxic related liver injury	Prednisone	Remission	[32]
F, 71	osteoarthritis	mRNA 1273 Moderna	4 days after dose I	Jaundice	ASMA 1:2560 (anti-actin)	Compatible with AIH	Steroids	Remission	[33]
M, 36	hypertension	ChAd OK1 nCOV-19 Astra Zeneca	26 days after dose I	Pruritus	ANA 1:160 speckled	Compatible with AIH	Steroids	Remission	[34]
F, 56	None	mRNA 1273 Moderna	6 weeks	Jaundice	ANA+ ASMA+	Compatible with AIH	Budesonide	Remission	[35]
F, 43	Dyslipidaemia	BNT 162b2 Pfizer BioNTech	15 days after dose I	Jaundice	Abs negative	Moderate portal infiltrate with interface hepatitis, biliary injury	Methyl-prednisolone	Remission	[36]

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
