# Peer review of "COVID-19 and Autoimmune Liver Diseases"

_jcm, 2022, doi:10.3390/jcm11102681_

Round 1
Reviewer 1 Report
Review 1 COVID-19 and autoimmune liver diseases
Annarosa Floreani 1,2,* and Sara De Martin 3
Suggestions: It should be more theoretical explantions for the occurance of autoimune diseases after SARS CoV2 infection.
Comments:
Check english language.
COVID-19 can active a hyperstimu- 50
lation of the immune system or, through the exposure to foreign peptides homologous to 51
human peptides (molecular mimicry), contribute to the development of autoantibodies 52
and autoimmune liver diseases.
Comment: activate
A list of numerous autoantibodies has been described as 53
linked to the SARS-CoV-2 infection, including ANA (anti-nuclear antibody), ANCA (anti-...
Comment: what means „linked“ – more prone to have COVID or more oftenly after COVID appears autoantibodies ?
As regard 55
the autoimmune conditions linked to COVID-19 infection, besides the olfactory manifes- 56
tations, a number of autoimmune diseases including SLE (systemic lupus erythematosus), 57
myasthenia gravis, and systemic sclerosis have been described [
Comment: Why olfactory manifestations were singled out
A total of 20,364 test-positive and 34,697 test- 82
negative subjects were analyzed. The authors found a negative association between auto- 83
immune status and risk of COVID-19 for tested-positive compared to tested-negative con- 84
trols.
Comment: “auto immune status“ is very wide, it should be more specific description.
A total of 254 patients with AIH were included, 92.1% of whom were 135
on treatment with glucocorticoids ...
Comment: It should be mentioned children or adults were studied.
All but one sub- 157
jects developed jaundice and marked alteration of liver enzymes. All but one developed 158
non- organ-specific autoantibodies.
Comment: “marked alteration of liver enzymes“. What kind of liver enzymes – all ? What about bilirubin ? How much ?
All patients were 163
treated with immunosuppressive therapy followed by remission.
Comment: What was the duration of remission ?
A limited number 201
of cases of AIH has been reported after vaccination against SARS-CoV-2, mainly with the 202
mRNA 1273 Moderna vaccine
Comment: This conclusion should be deleted, because very small numbers and no statistical difference between vaccines - five cases the trigger vaccine was the mRNA-1273 (Moderna), in three the PNT 162b2 Pfizer BioNTech.
Author Response
We are grateful for the review. Please consider our reply to your comments:
- Theoretical explanations for the occurrence of autoimmune disease after SARS-CoV-2 infection have been added.
- English has been revised.
- “active” has been changed to “activate”
- The development of autoantibodies in infected patients has been explained
- A comment on olfactory manifestations has been added.
- Results of the study performed in the open population assessing the association between autoimmune diseases and COVID-19 have been explained with more details (ref. 15).
- Age of population has been added (ref. 24)
- More clinical details, particularly on liver function tests of patients included in table 1 have been added.
- Duration of remission in not available.
- Conclusion has been deleted.
Reviewer 2 Report
In this review, the authors addressed the intriguing topic related to the impact of SARS-CoV-2 infection as a trigger of autoimmunity. They summarized the current knowledge about autoimmune liver diseases (AILDs) and COVID-19, focusing on 1) the risk of SARS-CoV-2 infection in patients affected by AILDs and/or under pharmacological treatment with immunosuppressants; 2) the capability of vaccination against SARS-CoV-2 to trigger autoimmune responses in the liver; 3) the efficacy of vaccines against SARS-CoV-2 in patients with AILDs. Results would support that the use of immunosuppressants in patients with AILDs seems to be correlated to COVID-19 severity. Moreover, few cases of autoimmune hepatitis (AIH) after SARS-CoV-2 vaccination have been also reported and all are characterized by a complete remission upon steroid treatment, but further evidence is needed to demonstrate the causality assessment.
They concluded that the link between SARS-CoV-2 infection and AILDs, presently, is far to be completely elucidated. In these patients, the use of immunosuppressants has been correlated to an increase of disease severity and lower levels of antibodies upon vaccination.
The manuscript is of interest and the topic of clinical impact. However, the link between autoimmunity and SARS-CoV-2 infection would need to be discussed in more detail. On one hand, the authors should recall some studies that reported the presence of ANA in Covid patients and, of more interest, ANA with the immunofluorescence "nucleolar" pattern (the serological marker of systemic sclerosis) in patients with severe lung disease, thus suggesting a prognostic significance of these antibodies in Covid disease, as previously reported (COVID-19 and Immunological Dysregulation: Can Autoantibodies be Useful? Clin Transl Sci. 2021 Mar;14(2):502-508).
On the other hand, the authors should report more details on the described cases of AIH in patients with SARS-CoV-2 infection and in patients who underwent vaccine.
Because this is a controversial topic, the authors should recall that autoimmune hepatitis is characterized by some specific serum autoantibodies. In particular, the most specific ANAs in autoimmune hepatitis are those with the "homogeneous" pattern as previously reported (Diagnosis and therapy of autoimmune hepatitis. Mini Rev Med Chem. 2009 Jun;9(7):847-60), while those more specific for PBC have other immunofluorescence patterns such as "membranous" or "multiple nuclear dots", as previously reported (Antinuclear antibodies giving the 'multiple nuclear dots' or the 'rim-like/membranous' patterns: diagnostic accuracy for primary biliary cirrhosis. Aliment Pharmacol Ther. 2006;24:1575-83; Antinuclear antibodies as ancillary markers in primary biliary cirrhosis. Expert Rev Mol Diagn. 2012;12:65-74).
-Regarding the described case of PBC/AIH overlap syndrome characterized by the positivity for AMA and anti-DNA antibodies, the authors should emphasize that such a concomitant serological positivity has been previously reported as the serological profile of the AIH/PBC overlap syndrome, thus confirming previous literature data (The serological profile of the autoimmune hepatitis/primary biliary cirrhosis overlap syndrome. Am J Gastroenterol. 2009;104:1420-5.).
-The last point that deserves to be further discussed is the one related to the age of patients who presented AIH after vaccine. Interestingly, as reported in Table 1, three patients were older than 70 years (76, 80, 71 years). The authors should recall that this is in line with previous literature data showing that AIH can (not rarely) also occur in the elderly as previously demonstrated (Clinical features of type 1 autoimmune hepatitis in elderly Italian patients. Aliment Pharmacol Ther. 2005;21:1273-7).
Author Response
We are grateful for the review. Please consider our reply to your comments:
- The link between autoimmunity and SARS-CoV-2 has been discussed in more detail.
- ANA test with the immunofluorescence nuclear pattern has been discussed and the relative reference has been added (Clin Transl Sci 2021) as suggested.
- More clinical details for patients reported in the table have been added.
- Discussion on ANAs in AIH and their specific pattern together with the specific autoantibodies for PBC has been added. Suggested references have been added as well (Rev Med Chem 2009; APT 2006; Exp Rev Mol Diagn 2012).
- A comment on anti-DNA antibodies and AIH/PBC overlap syndrome has been added. Three refs on this issue have been added: Clin Gastroenterol Hepatol 2014; AJG 2009; PLosOne 2018.
- A comment on the onset of AIH in the elderly has been added as suggested together with its relative ref: APT 2005.
Round 2
Reviewer 2 Report
The Authors satisfactorily addressed the raised points and the manuscript can be now accepted.